# Pacinian Corpuscles as a Diagnostic Clue of Ledderhose Disease—A Case Report and Mapping of Pacinian Corpuscles of the Sole

**DOI:** 10.3390/diagnostics12071705

**Published:** 2022-07-13

**Authors:** Jorge Feito, Ruth Esteban, María Lourdes García-Martínez, Francisco J. García-Alonso, Raquel Rodríguez-Martín, María Belén Rivas-Marcos, Juan L. Cobo, Benjamín Martín-Biedma, Manuel Lahoz, José A. Vega

**Affiliations:** 1Grupo SINPOS, Departamento de Morfología y Biología Celular, Universidad de Oviedo, 33003 Oviedo, Asturias, Spain; ruthesteban.m@gmail.com (R.E.); juancobodiaz@gmail.com (J.L.C.); javega@uniovi.es (J.A.V.); 2Servicio de Anatomía Patológica, Instituto de Investigación Biomédica de Salamanca, Complejo Asistencial Universitario de Salamanca, 37007 Salamanca, Spain; rrodriguezmarti@saludcastillayleon.es (R.R.-M.); mbrivas@saludcastillayleon.es (M.B.R.-M.); 3Servicio de Cirugía Plástica, Complejo Asistencial Universitario de Salamanca, 37007 Salamanca, Spain; mlourdes.garcia@usal.es; 4Servicio de Aparato Digestivo, Hospital Universitario Rio Hortega, 47012 Valladolid, Spain; fgarciaalo@saludcastillayleon.es; 5Departamento de Cirugía y Especialidades Médico-Quirúrgicas, Universidad de Santiago de Compostela, 15705 Santiago de Compostela, Spain; benjamin.martin@usc.es; 6Departamento de Anatomía e Histología, Universidad de Zaragoza, 50009 Zaragoza, Spain; mlahozg@unizar.es; 7Facultad de Ciencias de la Salud, Universidad Autónoma de Chile, Providencia 7500912, Región Metropolitana, Chile

**Keywords:** fibromatosis, Ledderhose disease, mapping, Pacinian corpuscles, immunohistochemistry, differential diagnosis

## Abstract

Background: Plantar fibromatosis, known as Ledderhose disease, is a neoplastic disease characterized by a locally-aggressive bland fibroblastic proliferation. Although Pacinian corpuscles alterations are commonly described in palmar fibromatosis, there are still no references about Pacinian corpuscles alterations in the rarer plantar version. Methods: We present a case report where a wide cutaneous resection, including the plantar fascia was performed, allowing a detailed study of Pacinian corpuscles. Pacinian corpuscles were analyzed using immunohistochemistry for neurofilament proteins, S100 protein, CD34, vimentin, glucose transporter 1, epithelial membrane antigen, neural-cell adhesion molecule, actin, desmin, type IV collagen, and high-affinity neurotrophin Trk-receptors. Moreover, the density and the size of the corpuscles were determined. Results: A clear increase in the number (hyperplasia) of Pacinian corpuscles was evidenced in the Ledderhose disease plantar fascia in comparison with similarly aged normal subjects. Pacinian hypertrophy was not demonstrated, but a significant decrease in the number of corpuscular lamellae was noted, with a subsequent increase in the interlamellar spaces. Pacinian corpuscles from the pathological plantar fascia showed an abnormal structure and immunohistochemical profile, generally without identifiable axons, and also absence of an inner core or an intermediate layer. Moreover, other molecules related with trophic maintenance of corpuscles were also absent. Finally, a vascular proliferation was commonly noted in some corpuscles, which involved all corpuscular constituents. Conclusion: The observed Pacinian corpuscles hyperplasia could be considered a diagnostic clue of plantar fibromatosis.

## 1. Introduction

Fibromatosis is a well-stablished soft tissue neoplasm characterized by a locally infiltrative fibroblastic proliferation [1]. Under the name fibromatosis are encompassed two separate clinical entities: palmar/plantar fibromatosis and desmoid-type fibromatosis [2,3]. In addition to the conventional WHO-approved names, these entities are also known as superficial (fascial) and deep (musculoaponeurotic or deep fascial) fibromatosis [1]. Although desmoid-type fibromatosis has been described in the foot [4], the WHO tumours of soft tissue and bone texts [2,3] do not consider these regions as possible location of desmoid fibromatosis. An anatomical basis for the existence of both variants of fibromatosis exists, as long as there is actually a superficial and a deep fascia in the foot, frequently fused [5]. However, the distinction between both neoplasms is nearly impossible, as both may have local recurrence if not completely excised, may display overlapping histological features, and lack metastatic ability [1].

Palmar/plantar fibromatosis was proposed to be a reactive process rather than a clonal neoplasm [6], histologically characterized by a proliferation of homogeneous immature-appearing spindled cells in a dense stroma [2]. Furthermore, fibromatosis was described to involve Pacinian corpuscles, and in this respect hyperplasia [7,8,9,10,11,12], hyperplasia/hypertrophy [8], hypotrophy [7], no changes or even absence of Pacinian corpuscles were all noted regarding palmar fibromatosis [10].

Superficial plantar fibromatosis is more infrequent than the palmar, which has made the study of this rare disease difficult. Thus, palmar fibromatosis affects up to 2% of general population, while plantar fibromatosis is included in the National Institute of Health’s list of rare diseases, with an imprecise incidence, affecting in any case <200,000 people [13,14], and we have found no studies of plantar fibromatosis referring changes in Pacinian corpuscles. From an anatomical standpoint, the foot shows an abundance of Pacinian corpuscles, a higher concentration was already noted in the weight-bearing area below the transverse metatarsal ligament [15,16,17,18]. The analysis of vibrotactile sensitivity supports this predominant localization of plantar Pacinian corpuscles around the thenar eminence [19,20]. These findings were confirmed in recent histological [21,22] and radiological [23] studies.

Here we analyze the Pacinian corpuscles of one case of plantar fibromatosis, initially diagnosed as a sarcoma. The differential diagnosis between fibromatosis and sarcoma may be difficult in both directions [1,3,24], and in the presented case the erroneous wide excision permitted the exhaustive study of Pacinian corpuscles. We compared the morphology and the immunohistochemical profile with corpuscles from normal soles, to check if plantar fibromatosis shares the Pacinian corpuscles alterations with the palmar version.

## 2. Materials and Methods

A 67-year-old woman visited the Plastic Surgery Department of Donostia University Hospital due to a plantar retraction, which was previously biopsied, with an initial histopathologic diagnosis of low grade leiomyosarcoma. After the initial diagnosis, a thickening of the plantar fascia was apparent on an MRI, concordant with plantar fasciitis. Exploration showed a retracted and a firm plantar surface, without ulceration or deformities. Finally, the patient was submitted to surgery, and a wide excision was performed due to the suspicion of a sarcoma. The final diagnosis of the patient in the resection specimen was plantar fibromatosis. During two years of follow up, no recurrences were found. The excised specimen measured 11.5 × 5 × 2 cm, while the cutaneous surface measured 10.5 × 2.5 cm. The piece was fixed in 10% formaldehyde and completely embedded in 2–3 mm thick paraffin blocks.

Two control cases, covering most of the plantar surface and including epidermis, dermis, hypodermis, and fascia, were also completely included in paraffin blocks to compare both the morphology and the density of the corpuscles. Controls came from wide tumoral resection pieces due to malignant neoplasms in the heel, corresponding to Donostia University Hospital and Salamanca University Hospital. The first control was a plantar skin sample of 9.5 × 4 cm and the second control was a plantar skin sample of 20.5 × 6.5 cm, including the five fingers; these samples did not have neoplastic involvement, neither macroscopically nor microscopically. Patients were 60- and 62-year-old males. These samples followed the same technical procedures as fibromatosis, sampling the whole pieces in paraffin blocks.

When the material was embedded in paraffin, several diagrams were made correlating the topography of each tissue block in order to constitute comprehensive maps of the plantar surface. Each paraffin block was cut in 5 and 20 µm thick consecutive sections, covering an approximate thickness of 100 µm; the first and the last slide were stained with hematoxylin-eosin (HE), leaving several blank slides (10 of 5 µm, and 2 of 20 µm thickness) to perform immunohistochemistry later on. Any identifiable corpuscle in the HE slides was counted, based on its onion-like lamellar structure, and included in the map of the sole of each case. Each corpuscle had its maximum diameter measured and its lamellae manually counted under photomicrographs. Corpuscles from the toes and the ball of the foot were excluded for comparison purposes. Photographs were taken with a Nikon Eclipse Ci microscope (Nikon Instruments, Tokyo, Japan), equipped with Plan Apo λ objectives (Nikon Instruments, Tokyo, Japan) and a MD-U3-10 microphotography camera (Catchbest Vision, Beijing, China).

The automated diagnostic platform Roche-VentanaTM Benchmark Ultra (Tucson, AZ, USA) was used for immunohistochemistry for diagnostic purposes and also in the 5 µm slides with identifiable sensory corpuscles. Antibodies used for diagnostic purposes of fibromatosis were both types of actin, desmin, S100 and CD34, among others. Antibodies were also directed against the main corpuscular constituents [25,26], and they have been used in formalin-fixed, paraffin-embedded human tissues (Table 1). The results of the fibromatosis case and the normal profile of Pacinian corpuscles is summarized in Table 2 in Section 3.2 of the Results.

## 3. Results

In the fibromatosis specimen, a widespread fibrous paucicellular proliferation was present, involving the dermis and, particularly, the hypodermis (Figure 1a), consistent with the diagnosis of plantar fibromatosis. The fibrous areas displayed positivity for vimentin and both clones of actin, with a very focal expression of desmin (data not shown).

### 3.1. Morphology of Fibromatosis and Control Corpuscles

Pacinian corpuscles were identified in all the cases investigated. Corpuscles from control specimens were located in the subcutis and surrounded by fatty tissue, with a preserved structure that was consistent with normality (Figure 1b).

Comparing the distribution between normal and pathological conditions, a striking tendency to form clusters in the fibrous areas of the fibromatosis case (Figure 1b) was apparent, while the control cases displayed a more uniform distribution along the plantar surface. Generally, corpuscles from the fibromatosis case were located in fibrous areas (Figure 1c,d). Moreover, the outer core capsular system was thicker with respect to a normal condition, and it showed unusual density and placement of capillaries (Figure 1e). In addition, the proliferation was accompanied by a striking nerve sheath fibrous thickening (Figure 1f).

### 3.2. Immunohistochemistry of Pacinian Corpuscles

The arrangement of various corpuscular constituents, including central axon, inner core, intermediate layer, and outer core capsular complex, was severely altered in almost all corpuscles (approximately 90%), with an absence of immunostaining for NFP (characteristic of the axon, Figure 2a,a’), S100 protein (characteristic of the inner core, Figure 2b,b’), vimentin (characteristic of the inner and outer cores, Figure 2c,c’), NCAM (characteristic of the inner core, Figure 2d,d’), type IV collagen (characteristic of the interlamellar spaces, Figure 2e,e’), and CD34 (characteristic of the intermediate layer, also demonstrates vessels, Figure 2f,g). In fact, while in Pacinian corpuscles from control cases, capillaries were restricted to the capsule, in those from the fibromatosis specimen, capillaries were also found in the outer core and even in the inner core space (Figure 2e–i and Figure 3). NTrk expression was neither detected (data not shown). Apart from these findings, the immunohistochemical profiles were consistent with normality (Figure 4 and Table 2).

### 3.3. Measurements of Pacinian Corpuscles

Assuming that the sampling included all the corpuscles, the fibromatosis case had 1.87 corpuscles per cm^2^, the first control had 0.5 corpuscles per cm^2^, and the second control had a density of 0.6 corpuscles per cm^2^, excluding the toes and the ball of the foot (Figure 5).

All the measured corpuscles showed a variable size and number of lamellae, consistent with tangential cutting of various corpuscles (Figure 5). The mean size of Pacinian corpuscles in control cases was 1242 (C.I. 968.75–1515.25) µm and 963 (C.I. 857.69–1068.31) µm, respectively, and the mean number of lamellae was 26.47 (C.I. 20.61–32.33) and 24.63 (C.I. 22.92–26.34), respectively. The mean size of Pacinian corpuscles in the fibromatosis case was 902 (C.I. 15.8–20.6) µm, ranging between 250 µm (the smallest corpuscle of the studied cases) and 3000 µm (the biggest corpuscle of the studied cases). The mean number of lamellae in this case was 18.2 (C.I. 95% 15.7–20.7), ranging between 5 (the corpuscle with less lamellae) and 47 (the corpuscle with most lamellae).

### 3.4. Statistical Analysis

Statistical comparison was performed through Student’s *t*-test, considering one group the 49 Pacinian corpuscles of the fibromatosis case and the other group the total 87 corpuscles in the controls. For the corpuscular size, the difference between both groups was 122 (C.I.−52–296) µm, with a *p* = 0.17. Regarding the lamellae count, the fibromatosis case had a mean of 18,2 (C.I. 95% 15.7–20.7), while the controls had a combined mean of 25 (C.I. 95% 23.2–26.9), meaning a difference of 6.8 (C.I. 3.8–9.9) lamellae between both, with a *p* < 0.001.

## 4. Discussion

Our observations regarding the distribution of Pacinian corpuscles in normal plantar fascia and in plantar fibromatosis are in good agreement with the normal anatomy, previously described with different approaches [15,16,17,18,21,22,23]. In plantar fibromatosis, Pacinian corpuscles morphologically showed hyperplasia and a significant diminution in the number of lamellae, but without a statistically relevant hypertrophy or hypotrophy. The increase in corpuscular density may be related with the progression stage of superficial fibromatosis, which may thereafter be followed by a more severe phase showing a decrease or an absence of Pacinian corpuscles [27,28].

As far as we know, publications reporting histologic evaluations of plantar fibromatosis did not mention Pacinian corpuscles [29,30,31]. In addition, specific reviews of plantar fibromatosis did not report nervous disarrangements [28,32]. Conversely, there is abundant information about changes in Pacinian corpuscles in palmar fibromatosis [7,8,9,10,11,12,27], and therefore the discussion of the present results must necessarily be in relation to this pathology. Moreover, a relationship between superficial fibromatosis and the peripheral nervous system has been described recently, noting diminished mean sensitivity in palmar fibromatosis patients [33].

As it occurs in the Pacinian corpuscles of palmar fibromatosis [12], we observed radical changes in the neuronal compartment of these plantar corpuscles, with disarrangement and absence of immunoreactivity for conventional markers of the elements contained therein. In particular, the generalized absence of an NFP-immunoreactive axon, S100 protein positive inner core and CD34-expressing intermediate layer clearly suggests corpuscular denervation, also found in a large number of palmar fibromatosis Pacinian corpuscles [12]. The other main structural finding in Pacinian corpuscles from plantar fibromatosis was the proliferation of capillary vessels involving the capsule, the outer core, and the inner core. This has not been reported earlier, and this finding might be consistent with the filling of the space left by the absence of the corpuscular neural compartment.

To explain the peripheral nervous system changes in palmar fibromatosis, a hypothesis was proposed based on increased nerve growth factor levels [8,9,34]. To explore whether neurotrophins could be involved in the changes of Pacinian corpuscles from plantar fibromatosis, the expression of the pan-neurotrophin receptor NTrk was studied. With the complete negativity observed for NTrk, the neurotrophin hypothesis seems not applicable for plantar fibromatosis. Other proposed mechanisms for Pacinian hyperplasia, which might be applicable in plantar fibromatosis, are related with microtraumatism-related perineurial fibrosis or with mastocyte-driven inflammation [7,8,9,10,11,12,35].

Thanks to the improvement on imaging techniques, Pacinian corpuscles and Pacinian corpuscle hyperplasia can now be diagnosed by MRI [22,36]. This attribute of Pacinian corpuscles hyperplasia can, thus, become a diagnostic clue of plantar fibromatosis, helping in the diagnosis of this entity in addition to other features [14,37,38].

## Figures and Tables

**Figure 1 diagnostics-12-01705-f001:**
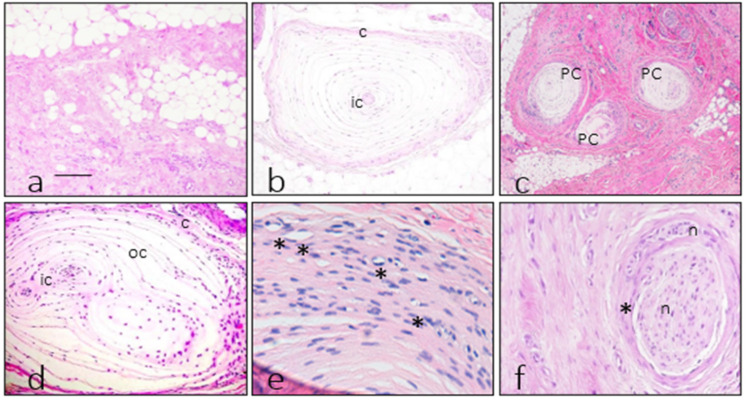
Morphology found in the fibromatosis and control cases. The fibromatosis case had a fibrous proliferation apparent in the subcutaneous tissue with low cellularity but the presence of infiltrative tracts in the hypodermal adipose tissue (**a**). Control Pacinian corpuscles (**b**) had a thin capsule (c), making contrast with the peripheral fatty tissue and also had a more colored inner core (ic) with a solid central structure with no nuclei consistent with an axon. The neoplastic fibrous proliferation encompassed some Pacinian corpuscles (pc), with a clustered appearance at low power (**c**). Corpuscles from the fibromatosis case (**d**) didn’t displayed the solid inner core/axon complex, and in the central region corresponding with the inner core area (ic) it was often apparent capillaries. The capsule of various corpuscles from the fibromatosis case (**e**) displayed a thickened capsule with an abundance of small blood vessels (*). Peripheral nerves (**f**) found inside the neoplastic fibrous proliferation showed intense fibrosis (*) entrapping nerve fascicles (n). Images (**e**,**f**) demonstrate the perineurial nature of the nerve fibers fibrosis employing consecutive slides showing a thick CD34 negative layer surrounding the nerve fiber (**e**) and the same thick layer positive for Glut1 (**f**). Scale bar correspond to 50 µm in images (**a**,**b**,**d**), 100 µm in (**c**), 20 µm in (**e**), and 30 µm in (**f**).

**Figure 2 diagnostics-12-01705-f002:**
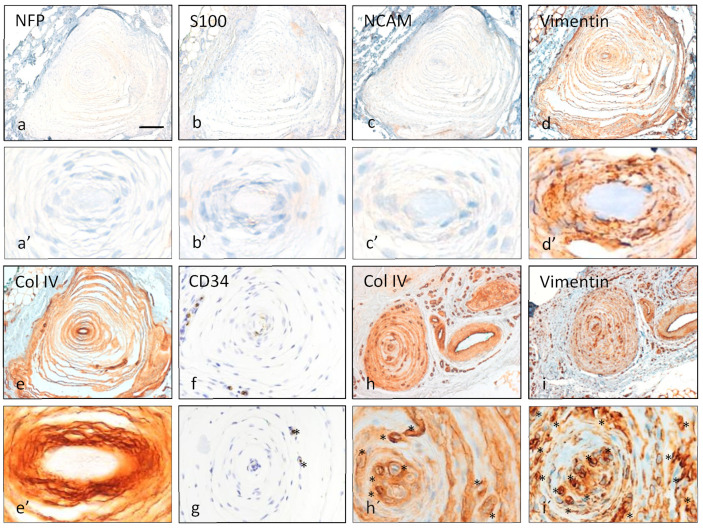
Immunohistochemical findings in consecutive sections of two different Pacinian corpuscles (**a**–**f**,**h**,**i**). The first corpuscle depicted (**a**–**f**) had no evidence of axonal profile innervating it due to an absent NFP expression ((**a**), detailed in (**a’**)); it had neither evidence of inner core, as long as there is no S100 ((**b**), detailed in (**b’**)) positivity and N-CAM was also negative ((**c**), detailed in (**c’**)); vimentin ((**d**), detailed in (**d’**)) left a central zone without expression in the region where the inner core should be. The outer core was well defined by type IV collagen ((**e**), detailed in (**e’**)). CD34 from different corpuscles was faintly present (**f**) or absent (**g**) surrounding the inner core region. The second corpuscle depicted in the bottom right (**h**,**i**) displayed a lamellar architecture of highlighted with Type IV collagen and vimentin expression; and a noticeable vascular proliferation (blood vessels highlighted with *) in the innermost and the capsular regions of the corpuscle was apparent in detail (**h’**,**i’**). Scale bar corresponds to 100 µm images (**a**–**e**,**h**,**i**), to 60 µm in images (**f**,**g**), 30 µm in the detailed images (**h’**,**i’**), and 15 µm in (**a**’–**e’**).

**Figure 3 diagnostics-12-01705-f003:**
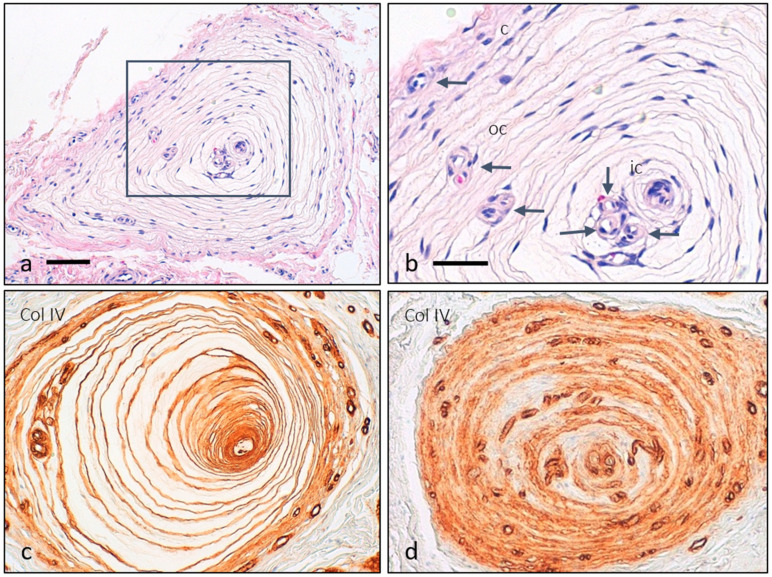
Capillaries in Pacinian corpuscles. The Pacinian corpuscle from Figure 2 is displayed with haematoxilin-eosin staining to identify the different corpuscular regions, including the capsule (c), outer core (oc), and inner core (ic); arrows indicate capillaries ((**a**), detailed in (**b**)). Typically, only the capsule of human Pacinian corpuscles contains capillaries, highlighted with type IV collagen immunostain in a typical control corpuscle, where only the peripheral capsule contains capillaries (**c**). Nevertheless, in those from fibromatosis disease, capillaries are present in all the corpuscular compartments (**d**).

**Figure 4 diagnostics-12-01705-f004:**
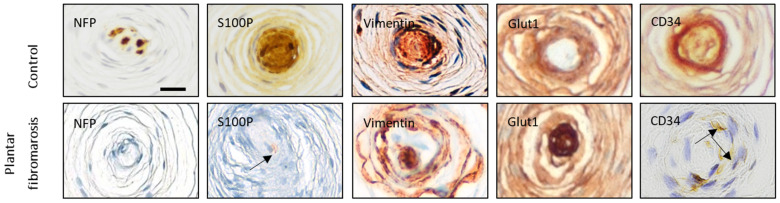
Representative images of the changes in the expression of some selected antigens in the Pacinian corpuscles of Ledderhose disease compared with controls. Scale bar = 25 µm.

**Figure 5 diagnostics-12-01705-f005:**
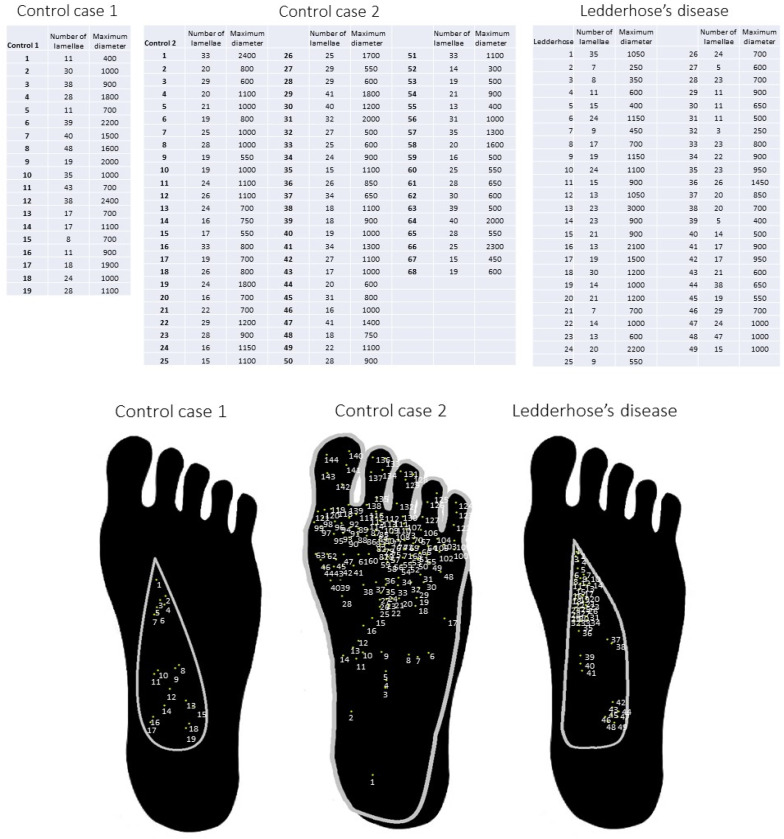
Schematic diagram of Pacinian corpuscle distribution in Ledderhose disease and in the control cases, including the measure and the lamellae count of each corpuscle. Total corpuscles in the first control case were numbered 1–19 for a total surface of 38 cm^2^. Total corpuscles in the second control case were numbered 1–68 for a total surface of 75 cm^2^, excluding the toes and the ball of the foot. Total corpuscles in the fibromatosis case were numbered 1–49 for a total surface of 26 cm^2^.

**Table 1 diagnostics-12-01705-t001:** Primary antibodies used in the study.

Antigen	Origin	Dilution	Supplier
Glut1 (polyclonal)	Rabbit	prediluted	Cell Marque ^1^
EMA (E29)	Mouse	prediluted	Cell Marque ^1^
N-CAM (123C3)	Mouse	prediluted	Ventana Medical Systems ^2^
NFP (2F11)	Mouse	prediluted	Cell Marque ^1^
S-100 protein (polyclonal)	Rabbit	prediluted	Ventana Medical Systems ^2^
Vimentin (V9)	Mouse	prediluted	Cell Marque ^1^
Actin, muscle specific (HHF35)	Mouse	prediluted	Ventana Medical Systems ^2^
Actin, smooth muscle (1A4)	Mouse	prediluted	Ventana Medical Systems ^2^
Desmin (D33)	Mouse	prediluted	Cell Marque ^1^
Collagen type IV (CIV22)	Mouse	prediluted	Ventana Medical Systems ^2^
CD34 (QBEnd/10)	Mouse	prediluted	Master Diagnóstica ^3^
NTrk (EPR17341)	Mouse	prediluted	Ventana Medical Systems ^2^

Glut1: glucose transporter 1; EMA: epithelial membrane antigen; N-CAM: neural-cell adhesion molecule; NFP: neurofilament protein; ^1^ Rocklin, California, USA; ^2^ Oro Valley, Arizona, USA; ^3^ Granada, Spain.

**Table 2 diagnostics-12-01705-t002:** Immunohistochemical profile of Pacinian corpuscles in normal plantar fascia (C) and plantar fibromatosis (PF). Quantification of the immunostain was performed considering absence of expression (−), faint expression (+), intermediate expression (++), and strong expression (+++).

Antigen		Axon	IC	IL	OC	C
NFP	Control	+++	−	−	−	−
	PF	− (1)	−	−	−	−
S100P	Control	−	+++	−	−	−
	PF	−	− (1)	−	−	−
EMA	Control	−	−	−	++	++
	PF	−	−	−	+/++	+/++
CD34	Control	−	−	+++	−	−
	PF	−	−	−/+	− (2)	− (2)
VIM	Control	−	++	+++	+++	+++
	PF	−	−/+++ (3)	+++	+++	+++
Actin	Control	−	−	−	−	−
	PF	−	−	−	−	−
Desmin	Control	−	−	−	−	−
	PF	−	−	−	−	−
IV COL	Control	−	++ (4)	−	−	−
	PF	−	−/+ (4)	−	−	−
Glut1	Control	−	−	−	+	+++
	PF	−	+++ (3)	−	+	++
N-CAM	Control	−	++	−	−	−
	PF	−	−	−	−	−
NTrk	Control	+	+	+	+	+
	PF	−	−	−	−	−

PF: Plantar Fibromatosis; C: capsule; IC: inner core, IL: intermediate layer; OC: outer core; (1) More than 90% of the corpuscles; (2) CD34 positivity in the endothelium; (3) less than 5% of the corpuscles; (4) pattern of basement membrane.

## Data Availability

Not applicable.

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
