# Peer review of "Pacinian Corpuscles as a Diagnostic Clue of Ledderhose Disease—A Case Report and Mapping of Pacinian Corpuscles of the Sole"

_diagnostics, 2022, doi:10.3390/diagnostics12071705_

Round 1

Reviewer 1 Report

This paper try to shed light about the Pacinian corpuscles alterations in plantar fibromatosis. The overall idea is interesting and can help the diagnosis of the disease and also a more targeted therapy, but in my opinion there are serious deficiencies in the work, that can not be accepted in this version. First of all the Authors can't arrive at any conclusion considering only 1 pathological sample. Furthermore the sarcoma is a tumor that can influence the tissue. Which is the real diagnosis, sarcoma or fibromatosis? It isn't clear. The Results are explained in a confusing way, the Figures and the quantification are not clear. It is fundamental to add at least on other sample and to be more precise in all the performed analysis following the suggestions below.

Abstract

- wide resection: explain also in the abstract the collection the sample. Resection of what?

- CD34 is repeated twice

- "This may be useful due to the rareness of the disease" Please explain. Useful for what?

Introduction

- also the deep fibromatosis involves fascia, ( the deep). In palmar and plantar fascia the superficial and deep fascia are fused. Pay attention describing this point

- rare disease: describe the incidence

- any clinical perspectives ?please describe this aspect.

Methods

- as mentioned before, 1 sample is not enough and the diagnosis is not clear

- 5 and 20 micron thickness? why? please explain

- Authors may indicate the dilution if the antibodies and describe the specificity of each antibody for Pacinian structures. Only in the Discussion this point is addressed. 

Results

- "abundant pacinian corpuscle": how many? please report also here the numbers

- Figure 1 in unclear: which is the control?

- the results lack of one Table or graph of quantification of the immunoreactions, indicated in Table 2. Did the Authors perform a quantification? Furthermore I'm not able to see any increase of capillary vessels in the pathological sample. 

Author Response

Thank you for your review. We performed a deep revision of several issues according reviewers suggestions and comments. In addition, we modified some linguistic and style issues. The real diagnosis is plantar fibromatosis, we revised the text to avoid such confusión with sarcoma, which was the erroneous initial diagnosis. Regarding your particular comments, we addressed all of them:

Abstract:

  • We added that the wide resection was from plantar fascia and skin.
  • The second CD34 was removed, we apologize for this mistake.
  • We removed the reference to the usefulness, leaving only the concept of diagnostic clue.

Introduction:                                                                                                                                                            

  • The mentioned issue refers to a pathological classification, described by the WHO and several books. The precise development of the disease is not perfectly established, but there are clinical differences between both pathological entities. In any case, the required precision was made, to include that deep fibromatosis may have origin also in the deep fascia.
  • The rareness of the disease was present in the 2 provided citations, but the required data about incidence was included in the manuscript, too.
  • The clinical perspective is mentioned in the discussion section to simplify this case report. In any case, the final paragraph of the introduction was enlarged to better introduce the case.

Methods:

  • This is an extraordinary case, as long as superficial fibromatoses are conservatively treated as a general rule. The reason of performing such a wide resection was the initial diagnosis of leiomiosarcoma. If you also consider the rareness of the disease, the case has an enormous value. Obtaining another case is not possible, as long as we are dealing with human patients here, and Please also take in consideration that there is a linving patient with partial amputation of the foot due to an erroneous diagnosis.
  • 20 µm slides were taken apart in the case a thick slide was needed for immunohistochemistry, and also aided to have different levels in the more easily usable 5 µm slides. We included the composition of the blank slides in the text.
  • The employed antibodies are all used in diagnosis of the Pathology Department with an automatized platform. These commercial antibodies are now served prediluted, so no dilution is required. However, there is actually a fault in the text, as long as the antibodies in the table also refer to the ones employed in the diagnosis of the fibromatosis disease. We included this point and also referenced the table of the results section.

Results:

  • We removed the mentioned “abundant” reference, and we, thus, leave to the section 3.3 of the results the judgement about the quantity of Pacinian corpuscles.
  • We changed section 3.1 about morphology of control corpuscles, as long as control corpuscles are already well described. We made again Figure 1 according to this approach, in order to make clearer this section.
  • Immunohistochemical results are complex, with different patterns of staining. The most relevant are depicted in Figure 2, but we don´t know how to report them in a graphic, so a table seemed the most feasible manner. We clarified the quantification in the legend of the table. We modified Figures 1 and 2 to better illustrate the capillary vessels increase. Also included Figure 3 for comparison.

Conclusions:

We removed this section.

Reviewer 2 Report

The authors performed a thorough analysis of Pacinian corpuscles in the plantar fascia of a case with plantar fibromatosis (Ledderhose´s disease) compared with two controls.

This case report is original and interesting. In fact, the authors report for the first time an increase of the number (hyperplasia) of Pacinian corpuscles along with abnormalities in their structure and immunohistochemical profile in Ledderhose disease plantar fascia.

However, the presentation of the study's results requires some improvements.

Data presented as supplementary figures are essential and should be better included in the manuscript as regular figures. 

The organization of figures 1 and 2 is not sufficiently clear for the readers. In particular, it is essential that these figures include images from the Ledderhose´s disease case vs. controls. For instance, there are no histological pictures showing hyperplasia of Pacinian corpuscles in the case vs. controls. The same is for the immunohistochemical data. Data from controls are shown only as semiquantitative scoring in Table 2. This is not sufficient. Figures 1 and 2 should include representative images of immunostaining for the different markers (at least, the most important ones) in Ledderhose´s disease case vs. controls.

Please carefully check the entire text for typos. 

Author Response

Thank you very much for your positive feedback and for the constructive review.

The conclusions section was removed, as it is a bit reiterative with the final paragraph of discussion. This final paragraph addressing the radiological findings was also modified to avoid confusion. We performed substantial changes in the manuscript according to other reviewer´s suggestions.

Reviewer 3 Report

The article discusses an interesting case of plantar fibromatosis with immunohistochemical study of the Pacinian corpuscles and final diagnostic considerations about the disease. The text sufficiently reviews the relevant literature, also in consideration of the rarity of the pathology. The quality of the histological and immunohistochemical images is truly appreciable, and the results are supported by a basic but well done statistical analysis.

In our opinion we suggest to improve the conclusions of the article, as they seem to contradict what is written in the discussion about the radiological findings.

Author Response

Thank you very much for your positive feedback and for the constructive review.

Figures 1 and 2 were revised according to your suggestions, also taking into account other reviewers; morphology images of control corpuscles were included in Figure 1 and immunohistochemical images comparing both conditions were included as Figure 3. Pacinian hyperplasia is based in the Pacinian measurements and statistical analysis, morphology is useful to describe architectural disarrangements. However, the clustering depicted in Figure 1 may serve as indication of the mentioned hyperplasia.

We also included the supplementary figures 1-3 as Figure 4 in section 3.3 of the Results.

We checked the text for typos, and we apologize for the reiterative inclusion of references as sequential numbers (intended), but also with the author and year of the citation (now removed).

Round 2

Reviewer 1 Report

The authors have significantly improved the work, making some points more clear. The new figure that compares the control and patient immunohistochemistry is appreciable. Now the diagnosis of the patient is also clearer, which was not absolutely clear in the first version: Authors can better explicit the preciousness of the sample.

Only 3 small comments, that need again to be improved:

1) the thickness of the specimens for IHC

2) the involvement of deep fascia in fibromatosis and the fusion of sup and deep fascia in the plantar region

3) the blood vessels indicated with the asterisks are not so clear, especially in Figure 2. Please add an inbox with bigger enlargement or a specific new Figure.

Author Response

Your review is highly appreciated. We have addressed your comments, and included some new text and references to highlight the rareness of the presented case. Moreover, we modified the title, to better reflect the clinical utility of this work.

  • The thickess of the IHC specimens is now mentioned in the last paragraph of the section, referred to IHQ.
  • The anatomy of the plantar fascia is not the focus of the article. As long as reviewer 1 insisted in this point, we included new text with bibliographic suppport in the introduction. We hope this new information results satisfactory.
  • We included a new figureto clarify the blood vessels changes, as suggested.

Reviewer 2 Report

All my previous comments have been addressed.

The quality of presentation and significance of content have been improved considerably.

Author Response

Thank you very much for your positive feedback and the support. We included a new versión to better address other reviewer´s indications. In the process, we even modified the title, to better reflect the clinical utility of this work.